# Risk Stratification of SARS-CoV-2 Breakthrough Infections Based on an Outbreak at a Student Festive Event

**DOI:** 10.3390/vaccines10030432

**Published:** 2022-03-11

**Authors:** Ralph Bertram, Vanessa Bartsch, Johanna Sodmann, Luca Hennig, Engin Müjde, Jonathan Stock, Vivienne Ruedig, Philipp Sodmann, Daniel Todt, Eike Steinmann, Wolfgang Hitzl, Joerg Steinmann

**Affiliations:** 1Institute of Clinical Hygiene, Medical Microbiology and Infectiology, Klinikum Nürnberg, Paracelsus Medical University, 90419 Nuremberg, Germany; ralph.bertram@klinikum-nuernberg.de; 2Study Program in Human Medicine, Paracelsus Medical University, 90419 Nuremberg, Germany; vanessa.bartsch@stud.pmu.ac.at (V.B.); johanna.sodmann@stud.pmu.ac.at (J.S.); luca.hennig@stud.pmu.ac.at (L.H.); engin.muejde@stud.pmu.ac.at (E.M.); jonathan.stock@stud.pmu.ac.at (J.S.); vivienne.ruedig@stud.pmu.ac.at (V.R.); 3Department of Internal Medicine II, University Hospital Würzburg, 97080 Wuerzburg, Germany; sodmann_p@ukw.de; 4Department of Molecular and Medical Virology, Ruhr-Universität Bochum, 44801 Bochum, Germany; daniel.todt@ruhr-uni-bochum.de (D.T.); eike.steinmann@ruhr-uni-bochum.de (E.S.); 5European Virus Bioinformatics Center (EVBC), 07743 Jena, Germany; 6Department of Research and Innovation Management, Biostatistics and Publication of Clinical Trial Studies, Paracelsus Medical University, 5020 Salzburg, Austria; wolfgang.hitzl@pmu.ac.at; 7Department of Ophthalmology and Optometry, Paracelsus Medical University Salzburg, 5020 Salzburg, Austria; 8Research Program Experimental Ophthalmology and Glaucoma Research, Paracelsus Medical University, 5020 Salzburg, Austria

**Keywords:** COVID-19, SARS-CoV-2, outbreak, breakthrough infection

## Abstract

In early 2022, the Coronavirus disease 2019 (COVID-19) remains a global challenge. COVID-19 is caused by an increasing number of variants of the Severe Acute Respiratory Syndrome Coronavirus 2 (SARS-CoV-2). Here, we report an outbreak of SARS-CoV-2 breakthrough infections related to a student festive event with 100 mostly vaccinated guests, which took place in Northern Bavaria, Germany, in October 2021. The data were obtained by retrospective guest interviews. In total, 95 students participated in the study, with 94 being fully vaccinated and 24 reporting infection by the delta variant. Correlation analyses among 15 examined variables revealed that time spent at the event, conversation with the supposed index person, and a homologous viral vector vaccination regime were significant risk factors for infection. Non-significant observations related to higher rates of infection included time since last vaccination, shared use of drinking vessels, and number of individual person-to-person contacts at the event. Our data suggest that a high rate of breakthrough infections with the delta variant occurs if no preventive measures are practiced. To limit infection risk, high-quality testing of participants should be considered a mandatory measure at gatherings, irrespective of the participants’ vaccination status.

## 1. Introduction

The severe acute respiratory syndrome coronavirus 2 (SARS-CoV-2) is a member of the betacoronaviruses, like the highly pathogenic Middle East respiratory syndrome coronavirus (MERS-CoV), and the first SARS-CoV [1]. Discovered in 2019 in patients with pneumonia in Wuhan, China [2], SARS-CoV-2 has been established as the etiologic agent of the Coronavirus disease 2019 (COVID-19) and was classified as a pandemic by the WHO on 11 March 2020 [3]. COVID-19 outcomes range from asymptomatic to multi-factorial, with symptoms including fever or chills, cough, shortness of breath, and loss of smell and taste. Infection can result in severe illness requiring hospitalization or even intensive care [4]. Worldwide, COVID-19 has a mean fatality rate of 1.5% [5]. SARS-CoV-2 is largely airborne transmitted by aerosols and droplets [6]. It has been reported that 1 L of exhaled breath by individuals who are infected contains a viral load of up to ~3000 SARS-CoV-2 particles [7], with an infectious dose being as low as 10 aerosol-borne viral particles [8]. Accordingly, non-pharmaceutical prevention interventions include medical-grade face masks, social distancing, ventilation of indoor spaces, and hand hygiene [9].

Safe and effective vaccines have game-changing potential against the COVID-19 pandemic. Vaccination is intended to induce B-cells that produce neutralizing antibodies against SARS-CoV-2 to prevent cellular infiltration. An additional effect is the stimulation of T-cells that provide cellular immunity and thus protect from the disease [10]. Furthermore, several studies provide evidence that vaccination is effective at preventing transmission [11]. When the current study took place, four vaccines against SARS-CoV-2 had been approved by the European Medicines Agency [12]. These were viral vectors (AZD1222, AstraZeneca, Cambridge, UK, and JNJ-78436735, Johnson&Johnson, New Brunswick, NJ, USA) or lipid-containing mRNA (BNT162b2, BioNTech/Pfizer, Mainz, Germany/New York, NY, USA and mRNA-1273, Moderna, Cambridge, MA, USA). The vaccines target the 1273 amino acids encompassing the spike protein of the SARS-CoV-2 Wuhan-Hu-1 (wildtype) variant, which is critical for viral entry into host cells [13]. The predominant variant of concern (VOC) in Germany in November 2021 was the SARS-CoV-2 variant B.1.617.2 (pangolin nomenclature, also known as the delta variant), which bears at least six characteristic spike mutations [14,15]. The delta variant shows increased transmissibility compared to the previously dominant VOC SARS-CoV-2 B.1.1.7 (alpha) [16].

Super spreader events are serious drivers of the COVID-19 pandemic [17,18,19,20,21,22] and breakthrough infections contribute to this effect [23]. Here, we report on an outbreak of SARS-CoV-2 delta in fully vaccinated students associated with a private festive event in October 2021 in Northern Bavaria, Germany. We examined a comprehensive set of data, based upon 15 variables obtained from 95% of guests. An extensive correlation analysis revealed that three factors were significantly associated with elevated risk of infection.

## 2. Materials and Methods

### 2.1. Description of the Outbreak Setting

The festive event took place on 21 October 2021, in Northern Bavaria, Germany, with a duration of about 8 h. The venue was a private setting, with a main room of approximately 85 m^2^ and 250 m^3^ of airspace (Figure 1). Assuming an equispaced distribution of all guests in the main room, the arithmetic mean of the area per person was 0.85 m^2^, corresponding to a personal radius of 0.52 m. Some party activities took place in a larger and less-frequented garage. To avoid noise pollution in the surrounding neighborhood, the windows in the main room were closed approximately 2 h after the start of the event.

### 2.2. General Information and Parameters of the Study

We conducted standardized telephone interviews with participants of the festive event who had given written consent. Data were collected in pseudonymized form, with individuals arbitrarily assigned unique numbers as identifiers. Table 1 summarizes the examined variables, risk factors, and symptoms. We defined 12 risk factors for severe progression of COVID-19 according to information provided by the Centers for Disease Control and Prevention (CDC) [24]. A case with a positive antigen point-of-care test (AgPOCT) or PCR-confirmed SARS-CoV-2 infection was defined as symptomatic if persons reported one or more symptoms according to the CDC [4] between 1 and 8 days after the event.

### 2.3. Contact Interaction Maps

Contact interaction maps for all guests and only those who tested positive were created by Bokeh (version 2.4.2) [25] based upon the questionnaire and supported by pictures taken at the event. Each guest was represented by a node: all interactions longer than 10 min and with a distance less than 1.5 m to another person were represented as edges between the nodes. Each node was started at a random position. The spacing on the diagram and the dimensions of the axes are arbitrary.

### 2.4. Statistics and Software

The data were compiled in Excel and checked for consistency. Crosstabulation tables were analyzed using Pearson’s Chi-Square test, Fisher’s Exact test, and the Kruskal–Wallis test for singly ordered tables. The Mann–Whitney U test and Bootstrap *t*-tests were used to test continuously distributed variables. Logistic regression analysis was applied to test and illustrate the effect of various risk factors on the risk of infection. All reported tests were two-sided, and *p*-values below 0.05 were considered statistically significant. All statistical analyses in this report were performed with STATISTICA 13 (Hill, T. & Lewicki, P. Statistics: Methods and Applications. StatSoft, Tulsa, OK, USA), PASW 24 (IBM SPSS Statistics for Windows, Version 21.0., Armonk, NY, USA), or GraphPad Prism (Version 9.3.1, GraphPad Software, San Diego, CA, USA).

### 2.5. Sequencing and Strain Assignment

The sequencing of SARS-CoV-2 isolates was performed on the Illumina platform using standard parameters. SARS-CoV-2 lineages were assigned using PANGO lineages [26,27]. Characteristic mutations were furthermore identified using outbreak.info [28]. Multiple sequence alignments were performed using QIAGEN CLC Genomics Workbench 21.0.5 (QIAGEN, Aarhus, Denmark). Sequences were deposited at gisaid.org [29] under the accession numbers: EPI_ISL_9107324, EPI_ISL_9107325, EPI_ISL_9107326, EPI_ISL_9107327, EPI_ISL_9107328, EPI_ISL_9107329, EPI_ISL_9107330, and EPI_ISL_9107331.

## 3. Results

The event took place in Northern Bavaria, Germany, in October 2021 with a duration of about 8 h in a poorly ventilated room of about 85 m^2^ and a connected garage (see Materials and Methods and Figure 1). Exactly 100 persons (all of them students) were at the location. Access was limited to those vaccinated, recovered, or who had tested negative for SARS-CoV-2 by either PCR or antigen point-of-care test (AgPOCT) prior to the event. Participants did not wear face masks or practice social distancing during the event. All further descriptions and calculations pertain to the 95 persons (95%) who provided data associated with the event, henceforth designated “participants”. Each one was given an arbitrary numeric identifier. Participants were 18–30 years of age: 53 males (56%) and 42 females (44%). Four of the participants had previously recovered from a COVID-19 infection. All of them had received 1–3 vaccinations post-infection. One individual was neither recovered nor vaccinated, but tested negative by AgPOCT on the day of the event. The other 90 individuals were vaccinated with 1–3 jabs with homologous and heterologous vaccination regimes (Table 2). The periods of the last administered vaccination ranged from 15 days to 269 days (mean: 142 days, median: 141 days) before the event. In the aftermath of the event, 24 of the participants (25%) were diagnosed as positive for SARS-CoV-2. All PCR-verified isolates were of the delta type and eight of them were sequenced. Seven were of the B.1.617.2 lineage and one was of the AY.122 (B.1.617.2.122) lineage. Eight persons, all fully vaccinated, reported COVID-19 symptoms at the beginning of the event. Two of them had tested negative by AgPOCT less than 24 hours prior to the event. Of the remaining six, person “1-2” was diagnosed positive by AgPOCT on day 1 and by PCR (cycle threshold = 24) on day 2 after the event. Thus, “1-2” was regarded as a probable index person. Selected data for individuals were infected are provided in Table 3. A proximity warning app [30] was used by 59 of the participants and gave a posteriori alert in 52 cases (88%). Fifteen of the positively tested had used the app and thirteen of them (87%) had received a warning.

Overall, 53 (56%) featured one or two risk factors for severe COVID-19. The participants attended the event for 60 to 510 min (mean 268 min, median 285 min). At least 89 guests were at the venue simultaneously with the supposed index person. Persons reported contacts with 4–80 other guests (mean 19.2, median 16). Figure 2, URL 1, and URL 2 provide static or interactive overviews of contacts among all participants (Figure 2a), and between only those who tested positive after the event (Figure 2b). In all, 36 participants reported conversing with the index person. None of the persons who were infected required hospitalization. However, 22 reported between 1 and 8 symptoms (Figure 3), with runny nose (n = 19), headache (n = 16), and loss of smell (n = 14) being most prevalent. One month after the event, 10 persons reported continued symptoms of decreased endurance or impaired sense of smell. 

In the search for risk factors for SARS-CoV-2 infection associated with the event, we analyzed possible correlations between 15 examined variables (see Methods section) and infection. Three correlations were statistically significant (*p* < 0.05):Extended stay at the event (Figure 4) (*t*-test, *p* = 0.0002). Notably, none of the guests that stayed 210 min or less tested positive.Conversation with the supposed index person. Of the 24 persons who were infected, 13 (54%) had talked to this person. This translates to a risk of infection of 36% compared to 17% without conversation (Fisher’s Exact, two-sided, *p* = 0.0497). For participants staying ≥3 h, we calculated risks of infection of 25% (without conversation) and 38% (with conversation).The risk of infection was significantly lower upon heterologous vaccination with AZD + BNT (23%) than with AZD + AZD (60%) (Fisher’s Exact, two-sided, *p* = 0.0489).In addition, the following non-significant observations were found (*p* ≥ 0.05):Risk of infection appeared decreased with BNT + BNT compared to AZD + AZD (Fisher’s Exact, two-sided, *p* = 0.14);Risk of infection appeared decreased with AZD + BNT compared to BNT + BNT (Fisher’s Exact, two-sided, *p* = 0.57);The time since last vaccination appeared to be inversely correlated with protection (logistic regression model, *p* = 0.38);Infections of guests who shared drinking vessels were slightly increased (Fisher’s Exact test, two-sided, *p* = 0.24);The number of contacts was slightly increased in subjects who were infected (21.6 ± 12.6) vs. subjects without infection (18.4 ± 12.6; generalized log-gamma model, two-sided, *p* = 0.18).

## 4. Discussion

A student festive event in Fall 2021 in Northern-Bavaria, Germany, emerged as a SARS-CoV-2 delta outbreak with a high rate of breakthrough infections. We retrospectively examined the spread of SARS-CoV-2 associated with the event. The primary infection rate was 25% among participants, of whom 96% were fully vaccinated. Accordingly, infection rates for the delta variant have been reported between 10.6% and 23.7% in a cohort with 96.2% vaccination rate during a nosocomial outbreak [31]. Extensive correlation analyses between 15 variables identified three significant factors for infection at the event.

First, extended stays at the event were a risk factor. Intuitively, airborne transmission is positively correlated to proximity and the time unprotected contacts spend with individuals who were infected [32]. This was also demonstrated at the study event.

Second, conversation with the supposed index person, in most cases closer than 1.5 m and for longer than 10 min, was a risk factor. This is in agreement with a study on younger (unvaccinated) students, in which proximity to a teacher who was infected and occasionally read aloud to the class while unmasked resulted in an almost three-fold increased primary infection rate [33].

Notably, among the 24 persons who were infected of our study, 10 declared no direct contact with the putative index person. Possible explanations for their infection are: (i) more than 1 index person was present (which may be supported by the finding of one other lineage among eight sequenced isolates); (ii) infectious particles spread through the air within the venue and, over time, accumulated above the infective threshold; and (iii) infection did not occur at the event.

Third, a vaccination scheme with two jabs of AZD compared to heterologous vaccination (with the 1st from AZD and the the 2nd from BNT) was a risk factor. In both cases, however, breakthrough infections occurred. Some vaccines against other infectious diseases practically provide lifelong protection and almost completely eradicate the etiologic agent upon encounter by a host. One example of this “sterilizing immunity” is the vaccine against measles, mumps, and rubella. Currently approved SARS-CoV-2 vaccines do not provide sterilizing immunity against COVID-19 [34]. This is consistent with reports on waned efficiencies of the approved vaccines in the delta surge [35,36,37]. Furthermore, vaccine effects against COVID-19 decrease with time after administration [38]. A three-shot vaccination regimen with BNT162b2 as a “booster” increases efficacy against delta [39,40], and accordingly, only one of the seven individuals (14%) with three doses of vaccine became infected in our study.

Our study is based on a very comprehensive set of data encompassing 15 variables, which permits multifold correlation analysis and delineation of recommendations. One limitation of the study is that the participants were rather homogeneous in terms of age and social status, which limits aspects of diversity. Another limitation is that only eight SARS-CoV-2 samples were available for sequencing and phylogenetic analyses. Therefore, we cannot exclude that additional positive index persons were present at the event.

## 5. Conclusions

This study shows that time spent at the event, conversation with the supposed index person, and a homologous viral vector vaccination regime were significant risk factors for SARS-CoV-2 breakthrough infection with the delta variant. While practicing social distancing, hand hygiene, and the use of face masks remain important measures in preventing the spread of SARS-CoV-2, high-quality testing of participants at in-person events could be considered mandatory to limit risk of infection, irrespective of the participants’ vaccination status. Further studies that include larger and more diverse cohorts and provide comprehensive datasets of several variables are required to shed further light on the interplay of risk factors for transmission of SARS-CoV-2 and its variants of concern—whether they are already known or have yet to emerge.

## Figures and Tables

**Figure 1 vaccines-10-00432-f001:**
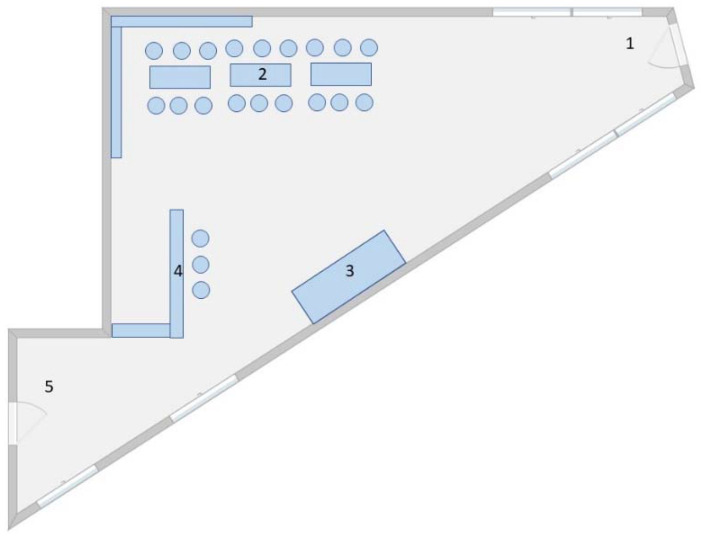
Floorplan of the venue in which the outbreak took place (only the main room is shown). Numerals represent the following: (1) entry area; (2) seating area; (3) sofa; (4) bar counter; (5) door to the garage. This figure is not drawn to scale.

**Figure 2 vaccines-10-00432-f002:**
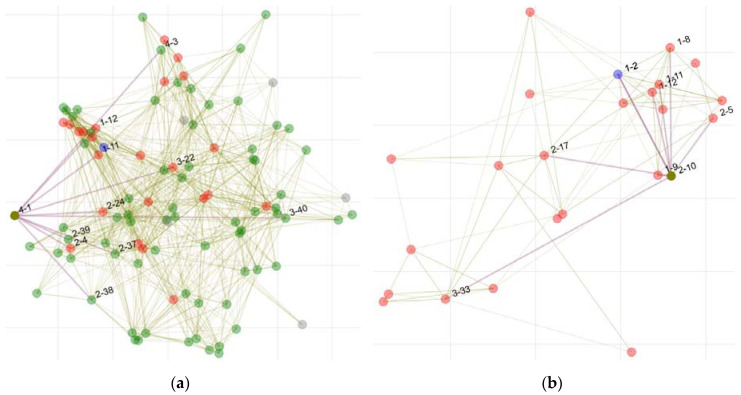
(**a**) Schematic overview of interactions between guests at the event (with individual “4-1” exemplary highlighted). Each guest is represented by a node, and all interactions longer than 10 min and with a distance less than 1.5 m are represented as edges between the nodes. The color of the nodes represents test results after the event: red, infection detected; green, no infection detected; dark green, selected individual for representation; grey, unknown. The blue node represents the putative index. An interactive version of the figure is available under Appendix A. (**b**) Contacts between individuals who tested positive after the event (with individual “2-10” exemplary highlighted). An interactive version of the figure is available under Appendix A.

**Figure 3 vaccines-10-00432-f003:**
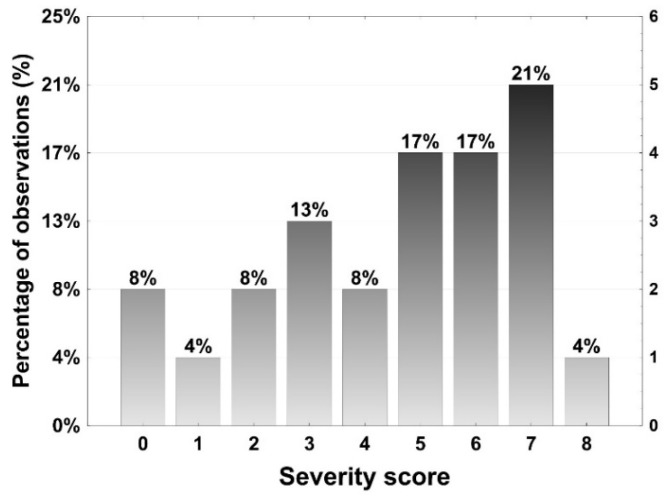
Histogram of severity of COVID-19 illness with participants who were infected, according to the number of symptoms.

**Figure 4 vaccines-10-00432-f004:**
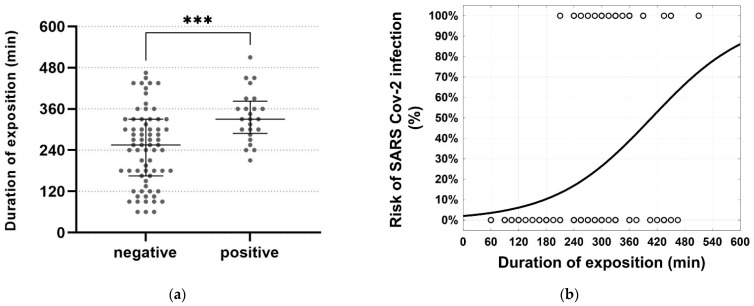
(**a**) Scatter dot plot of the time spent at the event in relation to a positive or negative test. The longer middle lines within the plots represent the median, the shorter lines within the plots bracket the interquartile ranges. Statistical analysis by *t*-test, two-sided, *** represents *p* ≤ 0.0002. (**b**) Calculated risk of infection relative to the duration of stay at the event.

**Table 1 vaccines-10-00432-t001:** Study variables and criteria.

Examined Variables	Risk Factors	Symptoms
Sex (male/female)	≥70 years of age	cough, fever (≥38.0 °C)
SARS-CoV-2-test before the event	male sex	common cold (blocked/runny nose)
COVID-19 symptoms on the day of the event	smoker	headache
COVID-19 vaccination status (including date of vaccination and type of vaccine)	(severe) obesity (body mass index > 30 kg/m^2^)	sore throat
Previously diagnosed SARS-CoV-2 infection	cardiovascular diseases	anosmia (loss of smell) or ageusia (loss of taste)
Risk factors (see middle margin)	chronic lung diseases	shortness of breath
Arrival and departure at the event	chronic kidney and liver diseases	gastrointestinal disorders
Contact to assumed index person (<1.5 m distance and >10 min or shorter/more distant conversation)	psychiatric disorders	myalgia
Individual contacts to other guests (<1.5 m distance and >10 min)	diabetes mellitus	chest pain
Shared use of drinking vessels	cancer	tachycardia
SARS-CoV-2 infection after the event (date and type of test)	immune suppression	
Symptoms up to 14 days after the event	trisomy 21	
Symptoms 1 month after the event		
Need for hospitalization		
Use and alert of Coronavirus warning app		

**Table 2 vaccines-10-00432-t002:** Vaccination status of participants.

Number of Participants	Previous SARS-CoV-2 Infection	1st Vaccination *	2nd Vaccination *	3rd Vaccination *
1	yes	JNJ	-	-
1	yes	BNT	-	-
1	yes	BNT	BNT	-
1	yes	BNT	BNT	BNT
1	no	-	-	-
1	no	JNJ	-	-
10	no	AZD	AZD	-
26	no	BNT	BNT	-
3	no	MOD	MOD	-
44	no	AZD	BNT	-
1	no	AZD	MOD	-
1	no	AZD	BNT	BNT
4	no	BNT	BNT	BNT

* AZD, AstraZeneca; BNT, BioNTech/Pfizer; JNJ, Johnson&Johnson; MOD, Moderna.

**Table 3 vaccines-10-00432-t003:** Selected data for individuals who were infected.

Individual	Vaccination Status ^#^	Time Since Last Vaccine Dose (d)	Number of Total Contacts	Time at the Event (min)	Type of Contact to Index Person *	Day of First Positive Test after Event	Ag-POCT	PCR (Cycle Threshold)	Number of Symptoms
1–2 (put. Index)	2 × BNT	178	12	270	n/a	+1	pos	pos (24)	7
1–6	2 × BNT	136	12	315	++	+4	pos	pos (29)	3
1–8	2 × BNT	192	11	255	++	+5	neg	pos (32)	7
1–9	2 × BNT	268	18	330	++	+2	pos	neg (36)	3
1–10	2 × BNT	112	18	450	++	+5	-	pos (19)	1
1–11	2 × BNT	154	33	360	++	+6	neg	pos (32)	7
1–12	1 × JNJ	157	19	330	++	+6	neg	pos (32)	0
1–13	2 × MOD	143	16	360	++	+5	neg	pos (27)	2
2–4	AZD + 2 × BNT	99	7	330	-	+4	pos	pos (20)	6
2–5	AZD + BNT	141	19	285	+	+6	neg	pos (30)	8
2–10	none	-	4	360	++	+6	neg	pos (24)	4
2–16	AZD + BNT	136	23	300	++	+4	pos	pos (24)	4
2–17	AZD + BNT	142	50	300	++	+4	pos	pos (20)	6
2–24	2 × AZD	156	52	360	++	+3	pos	pos (18)	5
2–27	AZD + BNT	141	19	360	++	+5	neg	pos (un-known)	6
3–2	AZD + BNT	136	26	345	-	+5	-	pos (17)	7
3–11	2 × AZD	165	20	330	-	+6	pos	pos (21)	3
3–22	AZD + BNT	157	38	510	-	+7	neg	pos (30)	0
3–26	AZD + BNT	127	16	210	-	+5	neg	pos (25)	5
3–27	AZD + BNT	141	11	240	-	+6	neg	pos (34)	6
3–28	AZD + BNT	141	8	240	-	+6	pos	pos (24)	2
3–30	AZD + BNT	141	33	435	-	+5	pos	-	5
3–33	AZD + BNT	141	28	390	-	+6	neg	pos (un-known)	7
3–34	AZD + BNT	148	25	450	-	+6	neg	pos (32)	5

^#^ AZD, AstraZeneca; BNT, BioNTech/Pfizer; JNJ, Johnson&Johnson; MOD, Moderna. * n/a: not applicable; ++: closer than 1.5 m (4.9 ft) and for longer than 10 min; +: conversation shorter and/or more distant; -: no direct contact.

## Data Availability

Viral sequences described in this study were deposited at GISAID under accession numbers: EPI_ISL_9107324, EPI_ISL_9107325, EPI_ISL_9107326, EPI_ISL_9107327, EPI_ISL_9107328, EPI_ISL_9107329, EPI_ISL_9107330, and EPI_ISL_9107331.

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
