# Peer review of "Risk Stratification of SARS-CoV-2 Breakthrough Infections Based on an Outbreak at a Student Festive Event"

_vaccines, 2022, doi:10.3390/vaccines10030432_

Round 1
Reviewer 1 Report
In this article by Ralph Bertram et colleagues, the authors analyzed the Risk stratification of SARS-CoV-2 breakthrough infections based upon an outbreak at a students' festive event.
Comments and suggestions:
- A summary diagram with all the steps of this complex study is recommended.
- Introduction section
Lines 44-45: there are many more symptoms than those listed. Revise it
Line 46: “COVID-19 has fatality rates of 46 around 1.5-2% in developed countries” - explain why the fatality rate was mentioned only for developed countries and not the rest of the countries.
Line 53: Explain the potential of vaccines on the pandemic. What about COVID-19 transmissibility or severity? But the influence on mutations? This phrase is ambiguous, revise it.
Discussion section:
-The novelty of the study and its clinical importance are not highlighted.
-Mention the limitations of this study.
-Line 229: define the statement "immune sterility", which is a term used by modern immunologists.
Also, compare the COVID-19 vaccines used in this study with a recognized vaccine that offers this sterile immunity mentioned by the authors.
Line 234: the authors state “SARS-CoV-2 and its variants of concern’ – I suggest adding the name of these variants of concerns of SARS-CoV-2.
The conclusion section is missing. Do the authors not have relevant conclusions for this research of theirs?
Consider revising accordingly!
Author Response
In this article by Ralph Bertram et colleagues, the authors analyzed the Risk stratification of SARS-CoV-2 breakthrough infections based upon an outbreak at a students' festive event.
Comments and suggestions:
- A summary diagram with all the steps of this complex study is recommended.
--> A figure with a more detailed description about data collection and data analysis has been added as supplementary material. By request, it may certainly be included into the main paper.
- Introduction section
Lines 44-45: there are many more symptoms than those listed. Revise it
--> We decided to list only the most prominent of the numerous COVID-19 symptoms (we therefore wrote “including”underscored by the phase “multi-factorial with symptoms including”). In order to account for long term effects, we have added another sentence to describe long-COVID symptoms. A more comprehensive list of symptoms is given in the Results section.
Line 46: “COVID-19 has fatality rates of 46 around 1.5-2% in developed countries” - explain why the fatality rate was mentioned only for developed countries and not the rest of the countries.
--> We like to excuse for not drawing further countries into account and have changed the sentence to provide a worldwide perspective.
Line 53: Explain the potential of vaccines on the pandemic. What about COVID-19 transmissibility or severity? But the influence on mutations? This phrase is ambiguous, revise it.
--> We thank the reviewer for this suggestion. We have added two more sentences to address the effects of vaccination on the severity of disease and on transmission in the Introduction section. The efficiency of vaccines against SARS-CoV-2 variants of concern is addressed in the Discussion section, as it pertains to our data as well.
Discussion section:
-The novelty of the study and its clinical importance are not highlighted.
-Mention the limitations of this study.
--> Strengths and limitations of our study are now described in the Discussions section, with the novelty of our work now also clarified in the Introduction section.
-Line 229: define the statement "immune sterility", which is a term used by modern immunologists.
Also, compare the COVID-19 vaccines used in this study with a recognized vaccine that offers this sterile immunity mentioned by the authors.
--> The term “sterilizing immunity” is now explained and provided with an example. We would prefer to omit quantitative comparisons of vaccines’ efficiencies against different diseases because it does not necessarily provide information about the duration of protection.
Line 234: the authors state “SARS-CoV-2 and its variants of concern’ – I suggest adding the name of these variants of concerns of SARS-CoV-2.
--> Those variants of concern that were encountered most frequently in the past are listed in the Introduction section. The statement in the Conclusions section rather relates to VOCs possibly emerging in the future. We have added text to clarify.
The conclusion section is missing. Do the authors not have relevant conclusions for this research of theirs?
Consider revising accordingly!
--> We thank the reviewer for this comment. A Conclusions section that includes further aspects and recommendations was added.
Reviewer 2 Report
The manuscript addresses an interesting and up-to-date topic, that is, the risk stratification of SARS-CoV-2 breakthrough infections. It is very well structured and presented, and it uses an adequate methodological design according to the aims of the research. Nevertheless, before publication, there are some flaws that need to be overcome, namely:
(1) the previous empirical studies used in the introductory item need to be expanded, in order to reinforce the supportive background of the current study, including two previous studies from your target journal;
(2) the innovative contribution needs to be outlined, in the introductory item;
(3) the limitation concerning the use of a students' sample needs to be incorporated into the analysis, in the concluding remarks; and
(4) the implications derived from the current study, should support recommendations for public health policy, to be also incorporated in the concluding remarks.
Author Response
The manuscript addresses an interesting and up-to-date topic, that is, the risk stratification of SARS-CoV-2 breakthrough infections. It is very well structured and presented, and it uses an adequate methodological design according to the aims of the research. Nevertheless, before publication, there are some flaws that need to be overcome, namely:
(1) the previous empirical studies used in the introductory item need to be expanded, in order to reinforce the supportive background of the current study, including two previous studies from your target journal;
--> As suggested two more relating studies were added to the Introduction section in support of our current study.
(2) the innovative contribution needs to be outlined, in the introductory item;
--> We now state the novelty of our study more clearly at the end of the Introduction section.
(3) the limitation concerning the use of a students' sample needs to be incorporated into the analysis, in the concluding remarks; and
--> Strengths and limitations of our study are now described in the Discussion section.
(4) the implications derived from the current study, should support recommendations for public health policy, to be also incorporated in the concluding remarks.
--> In the Conclusions section, we now state our recommendations more clearly.
Round 2
Reviewer 1 Report
The MS is revised, I have only a last suggestion before acceptance: The figure made is unattractive, arrows should be included; also the diagram should be included in the main manuscript.